# Securing Flows in the River Systems through Irrigation Water Use Efficiency—*A Case Study from Karula River in the Ganga River System*

**Nitin Kaushal [1,*], Suresh Babu [1], Arjit Mishra [1], Rajesh Bajpai [1], Phanish Kumar Sinha [2], Rama Kant Arya [3], David Tickner [4] and Conor Linstead [4]**

[1] World Wide Fund for Nature—India (WWF India), New Delhi 110003, India
[2] Water Resources & Irrigation Expert and Partner WWF India, Lucknow 226001, India
[3] Irrigation Expert and Partner WWF India, Lucknow 226001, India
[4] World Wide Fund for Nature UK, London GU21 4LL, UK
[*] Correspondence: nkaushal@wwfindia.net

**Abstract:** The pressure on freshwater resources is leading to diminishing flows in some of the critical river systems across the globe. India is no exception, and this is mainly because of water withdrawal for irrigation, which is often to the tune of 70% to 80% of the lean season flows, with some proportion for domestic and industrial use. While graduating from the concept of environmental flows and its assessment methodologies in India, the water-managers, the researchers and the conservationists are now moving towards answering the next question, if the rivers are to be revived, where will the water come from, especially in the case of over-allocated rivers, including the River Ganga. While the logical way is to look at the biggest user of water, i.e., irrigation, it remains to be seen whether the irrigation water savings will actually lead to enhancing flows in a river, complementing the efforts towards maintaining e-flows in rivers, or whether it will lead to more area under agriculture, bring changes in cropping patterns towards more water-intensive crops or result in something else. This is a growing debate across the globe, where India is no exception, and there has been a wide range of opinions in this regard. This paper discusses the process, findings and lessons from a joint initiative involving farmers, the Uttar Pradesh state Irrigation and Water Resources Department, Bijnor District Administration and a conservation organisation, WWF, to enhance flows in a sub-tributary, called the Karula River, which is part of the Ganga River system. Another objective of this paper is to look at the scalability and replicability of similar approaches in other irrigation command areas to benefit nearby river systems in general. Under this initiative, the team attempted to enhance flows in the river Karula by routing the saved water from irrigation supplies in a canal commanded area. This saving of water is being achieved due to supply-side and demand-side measures that are being adopted in the project area. With the objective of ensuring the sustainability of the initiative, efforts are made to form an institutional arrangement, through which this initiative can be sustained beyond the project support.

**Keywords:** irrigation water use efficiency; environmental flows; river conservation; Water Users Association; minor canal; Ganga; Ramganga

## 1. Introduction and Context

Rivers, wetlands and aquifers are a critical source of water for nature, biodiversity and human beings. In fact, these sources have their own inter-dependent ecosystems. All of these ecosystems face multiple challenges, in the wake of:

a. feeding a growing population in a changing climate, while also conserving and restoring nature

b. reconciling multiple competing human demands for water, further compounded by changing lifestyles, market-driven processes and unplanned developmental activities

c.　　ensuring sustainable water use, in line with Sustainable Development Goal 6 (SDG 6) [1] which calls for "ensuring availability and sustainable management of water and sanitation for all"

Grill et al. 2015 [2] concluded that, globally, 48% of river volume is moderately to severely impacted by either flow regulation, fragmentation, or both. This situation calls for maintaining or restoring flow regimes, in the form of environmental flows, to ensure the maintenance of ecological integrity. The most referred definition of e-flows by Arthington et al., 2018 [3] is "the quantity, timing, and quality of freshwater flows and levels necessary to sustain aquatic ecosystems which, in turn, support human cultures, economies, sustainable livelihoods, and well-being".

The inclusion of environmental flows in IWRM (Integrated Water Resources Management) is likely to result in increased effectiveness of environmental outcomes along with many benefits to social well-being and economic return, Hirji and Davis 2009 [4] Environmental flows can form the basis for an integrated approach to water allocation and river operation. Identifying environmental flows is likely to provide a strong scientific and open process within river management and for water allocation decisions at a basin scale, Overton et al., 2014 [5].

Currently, 41% of global irrigation water use occurs at the expense of e-flows requirements and India contributes to about 17.7% of global annual e-flows deficit (both in terms of the total annual deficit and the number of months with transgressions) Jägermeyr et al. 2017 [6].

Excessive use of surface-water and ground-water for irrigation has led to a diminishing water-table and the transformation of perennial rivers into seasonal ones. Stockle 2002 [7] noted that withdrawing surface water implies changes to the natural hydrology of rivers and water streams, affecting the aquatic ecosystems associated with these water bodies.

The river basins in South Asia (including the Ganga basin), in the Mediterranean region, and the Sahel are most sensitive to irrigation improvements resulting from the combination of local crop types, climate and soil conditions and the current irrigation system. India has 18% of world population, having 4% of world's freshwater, of which 80% is used in agriculture, Dhawan 2017 [8]. In the Indian context, the concept of e-flows is in the process of being mainstreamed in river basin management. However, barring a few exceptions, efforts have been largely centered around the Ganga River, within which the focus has been on developing the technical foundations for e-flows assessment. Efforts are also being made to understand the tradeoffs, in cases where e-flows are to be maintained. However, the implementation of e-flows remains elusive and there is a particular need for practical case studies documenting how irrigation management can aid maintaining e-flows.

Appendix A provides brief account of E-Flows assessment and implementation in rivers in India, this listing is developed from Brij Gopal 2013 [9].

Tickner et al., 2020 [10] pointed out that case studies of environmental flows implementation, successful or otherwise, provide valuable insights into barriers and enabling factors, and illustrate the evolution and propagation of the practice of environmental flows globally. Kaushal et al., 2019 [11] documented approaches to understand and resolve potential trade-offs between environmental flows objectives for the Ganga River in Uttar Pradesh and agricultural water demand. They concluded that, contrary to common perceptions, the increase in water needed to restore flows is likely to be small, compared to overall water demand. Moreover, the implementation of irrigation water use efficiency measures can ameliorate the potential adverse impact on farmers from changes in water allocation.

The National Commission for Integrated Water Resources Development, Government of India had estimated total withdrawal/utilisation for 2010 for all types of uses as 710 BCM (Billion Cubic Meters) in a high projection scenario. Of this, irrigation accounted for nearly 78%, followed by domestic use of 6%, industries at 5%, power development at 3%, and other activities claimed about 8% including evaporation losses, and environment and navigational requirements, CWC 2020 [12]. With this background, it becomes imperative to

engage with the irrigation and agriculture sector around water use efficiency, if freshwater resources (rivers, lakes and wetlands) are to be conserved.

This paper reports on the process and lessons from an initiative to enhance flows in the Karula river, The basic premise of this work is an ask—can we help secure e-flows in the river, through interventions in the irrigation sector, while maintaining sustainable and enhanced water and land productivity levels, with improved overall agricultural production? While the logical way is to look at the biggest user of water, i.e., irrigation, it remains to be seen whether the irrigation water savings will actually lead to enhancing flows in a river, complementing the efforts towards maintaining e-flows in rivers, or whether it will lead to more area under agriculture, bring changes in cropping patterns towards more water-intensive crops or result in something else. This paper reports on the process and lessons from an initiative that ran from 2017–2021 to enhance flows in Karula River through the implementation of supply-side and demand-side measures in the Khanpur Minor canal command area in Bijnor district of Uttar Pradesh in India.

*Project Area*

Under the Karula river pilot project, the aspiration has been to enhance the diminishing flows in the Karula river, a tributary of the Ramganga River system (itself a tributary of the Ganga), from the saved water from the irrigated command area of a minor canal, called Khanpur Minor. The catchment area of Karula river is 957 km$^2$, which is little over 4% of the catchment area of Ramganga basin (25,028 km$^2$). This canal system is operated and maintained by Uttar Pradesh Irrigation & Water Resources Department (UPI&WRD). The land use and land cover of Karula river catchment is agriculture dominated and the irrigated command area of Khanpur Minor canal is reflective of the same. The land-use and land-cover mapping for both Khanpur Minor command area and Karula River catchment is available as Appendices B and C.

The Karula River is not gauged, therefore the team started taking hydrological and hydraulic observations on fortnightly basis since mid of 2017 (with an objective to develop some understanding of flows in the river). The information on flows and hydraulic parameters were generated through these on-site observations (2017–2021). This led to the understanding of the general flow's patterns and variations in water levels in the river across various seasons. Based on four years of data, the dependability curve was developed and the same is presented in Figure 1.

Ramganga water resources are stored in a reservoir called Kalagarh Dam, which is the second biggest dam-based reservoir (after Tehri dam) in the state of Uttarakhand, storing over 2448 million cubic meters of water, as per the National Register of Large Dams 2019, Central Water Commission [13]. The Kalagarh Multi-purpose Project was designed for irrigation, flood protection and production of electricity with an installed capacity of 198 MW (Uttarakhand Irrigation Department). The major proportion of water in Kalagarh dam is allocated to augment the lower Ganga Canal System (85%) while the rest is allocated to small independent canal systems known as the Ramganga sub feeder (10%) and Pheeka canal system (5%)—as per the information from authorities.

As a result of the diversion of Ramganga waters for irrigation canals, the flows in the Ramganga downstream barrage (Hareoli Barrage) are miniscule for the middle stretch of the Ramganga river. The lower stretch of the Ramganga, however, just before joining the Ganga, is relatively better due to contributions from the tributaries in the middle to lower stretches of the Ramganga River.

The Ramganga sub feeder canal system takes off from Kho Barrage built on the Kho River in Bijnor district (see Figure 2). This Ramganga sub feeder main canal has a series of minor canal systems extending irrigation supplies to the farms in three districts of Uttar Pradesh; one such minor canal is called the Khanpur Minor Canal, having a designed discharge of over 0.01 cubic m/s (3.5 cubic feet/second or 100 L/s). The irrigation command area of this canal system largely falls in four villages (Khanpur, Meerapur,

Rehtoli, Kolasagar) of Seohara Block in Bijnor district of Uttar Pradesh. Figure 2 illustrates the location of the pilot area on the map of the country.

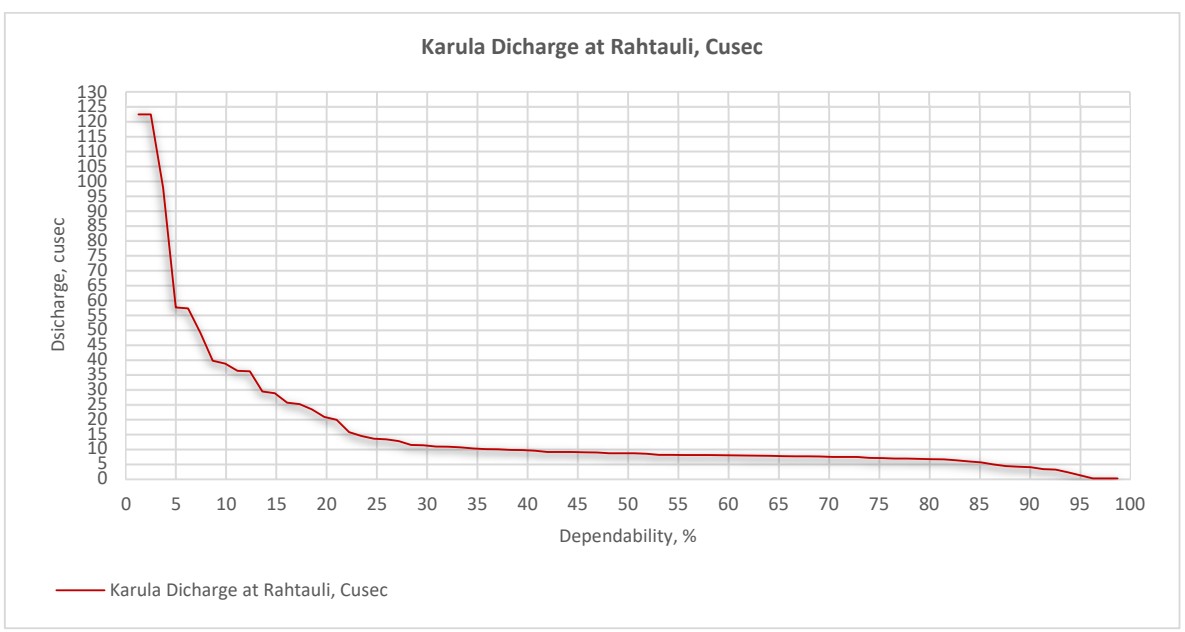

**Figure 1.** The Flows Dependability Curve of Observed Discharges in Karula River at Rahtauli village point and hydrological variations in the Karula River from 2017 to 2021.

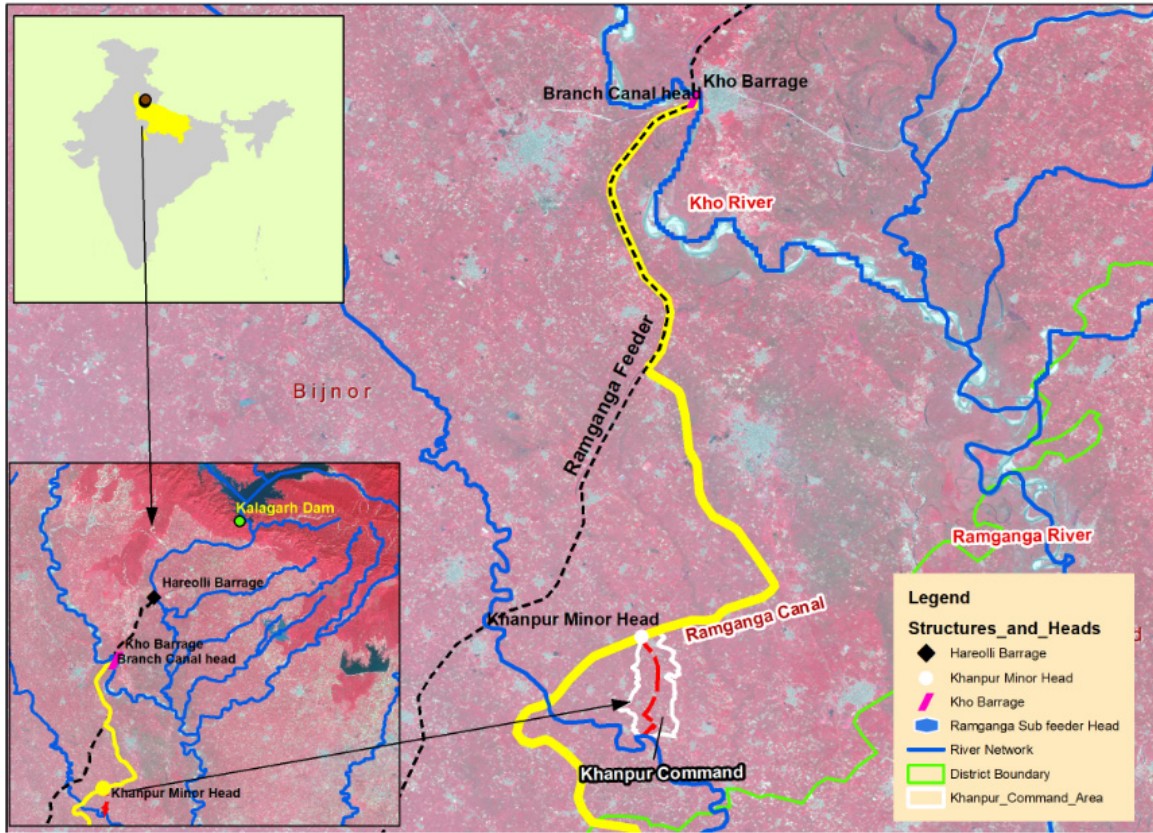

**Figure 2.** Location map of the Karula pilot area.

The farmers in the catchment of the Karula river predominantly grow sugarcane, not only because of rich water resources and the presence of sugar-mills in the nearby

areas (Seohara and Dhampur), but also due to the high economic value of sugarcane, as a cash crop, and the prevailing Minimum Support Price, which attracts farmers for assured incomes. According to a broad estimate, about 67% of the Khanpur Minor command area, i.e., about 260 ha, grows sugarcane and on the rest of the command, the usual wheat-paddy is grown. (Landuse Map, WWF-India—available as Appendix B)

The key statistics of the Khanpur Minor Canal are tabulated in Table 1.

**Table 1.** The main features of the Khanpur Minor Canal System, including CCA (Culturable Command Area, the area which can be physically irrigated from a scheme and is fit for cultivation) and PPA (Proposed Protected Area, the area that is assured for irrigation by a scheme).

| S. No. | Item | |
|---|---|---|
| 1 | Length of Khanpur Minor Canal | About 3 km |
| 2 | CCA (Culturable Command Area)<br>PPA (Proposed Protected Area)<br>a. Rabi (Cropping season from July to October)<br>b. Kharif (cropping season from November to March/April) | 389 Hectare<br>148 ha<br>124 ha |
| 3 | Number of Farmers | 311 |
| 4 | Passage to connect tail-end of Canal with the nearest Karula river-bank (constructed as part of this initiative) | Over 554 m |

The tail end of the Khanpur Minor canal system is about 554 m (acceptable route) from the left bank of the river Karula. Therefore, one of the tasks under this initiative was to construct a passage to connect the tail-end of the canal with the left bank of the Karula river. This passage is a mix of open earthen and lined channel, with some portion as underground-pipeline.

## 2. Approach and Methods

### 2.1. Context and Approach

The idea of the Karula pilot has been conceived keeping in view a stakeholder-centric participative approach, wherein the farmers, concerned state government institutions (especially the Uttar Pradesh Irrigation and Water Resources Department) and district authority (Bijnor district) were key stakeholders. Whilst the project team led and coordinated the entire task, the stakeholders, local knowledge and wisdom played a critical role, in terms of contextual guidance, rapport building and farmer-level coordination.

A three-pronged approach was adopted to implement the pilot activities, including:

a. Demand-Side Management (promotion, demonstration and adoption of irrigation water used in efficient ways and means, in terms of Better Management Practices, to save water)

b. Supply-Side Management (rehabilitation of the entire canal system of Khanpur Minor, including the construction of a passage from the tail-end of Minor to the riverbank of Karula)

c. Institutional Strengthening (facilitation of the constitution of the Khanpur Minor Water Users Association and capacity building of command farmers to make them well-acquainted with various key provisions of the Uttar Pradesh Participatory Irrigation Management Act, 2009—under which the Water Users Associations are formed in the state)

Key amongst the above three aspects of the approach has been the inclusion of socio-economic aspects, technical considerations, and stakeholder engagement. During implementation of the three-pronged approach, these aspects were not only taken into account, but were of central focus.

The Karula river initiative began with the assessment of baseline information pertaining to farmers, their landholdings, literacy rate, cropping cycle and cropping pattern,

modes of irrigation, agricultural yield, input cost, profit margins, the status of canals and allied infrastructure, and more. With the increased understanding about the area, the work began, wherein the role of various stakeholders (including command farmers, Uttar Pradesh Irrigation and Water Resources Department (UPI & WRD), Bijnor District Administration and WWF India) was critical.

*2.2. Stakeholders Engagement*

Stakeholder engagement is seen as a means of contributing to improved water governance, where governance is defined as the policy and practices giving rise to particular forms of water management in different contexts (Wehn et al., 2018) [14]. Various stakeholders under the Karula initiative had played an inclusive and iterative part in realising the larger objective. Although their responsibilities were distinct with overlapping roles, they did appreciate each other's contribution and collaborated to work for the larger water conservation goal. For instance, the supply side-interventions (rehabilitation and maintenance) on the Khanpur Minor canal are a Uttar Pradesh Irrigation and Water Resources Department task, but farmers and other stakeholders played a critical role in the overall supervision and coordination.

On the other hand, a passage was required to be constructed to connect the canal's tail-end with the riverbank, which was purely physical activity. Here, the technical guidance of the Uttar Pradesh Irrigation and Water Resources Department was obtained, yet the farmers played the key role, as they deliberated and finalised the alignment of the passage route. In this process, the involvement of district authorities was critical to provide information about the rights (based on revenue records) on the land between the passage routes. Only then could all stakeholders take the final call on the passage route and the work begin.

Social learning has been an added advantage of such stakeholder-centric approaches. One of the most salient aspects of social learning is the collective—rather than individual—process of learning, knowledge co-creation and accumulation of wide experiences to generate a broader knowledge and evidence base, from which decisions can be taken (Wehn et al., 2018) [14]. In terms of the Karula initiative, it has been a mutual learning for all stakeholders. For instance, whilst the team promoted trench-based sugarcane farming in the Khanpur Minor canal command, farmers came up with the idea of multi-cropping by making use of moisture in the soil, and therefore growing other crops to maximise their economic gains. Some of the progressive farmers in the adjoining areas as well as the Department of Sugarcane, Government of Uttar Pradesh (*Success Stories of Sugarcane development in District-Bijnor, Uttar Pradesh, 2018* [15]) were promoting these practices. As a result of knowledge exchange, exposure visits and personal initiatives, many farmers adopted this idea. This has also been a learning experience for their fellow farmers (even outside the command area).

The chronology of the stakeholders' engagement process is illustrated in Figure 3.

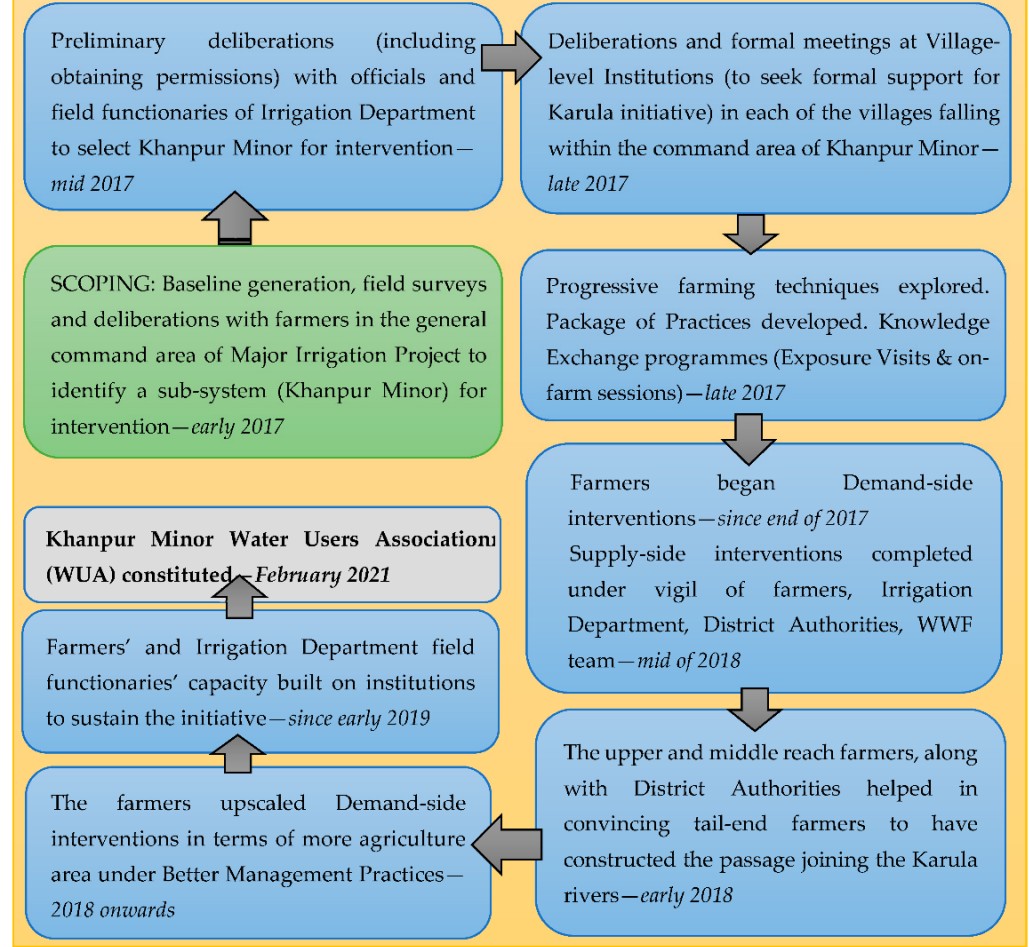

**Figure 3.** Stakeholder's engagement, along with timelines and key steps and milestones.

*2.3. Role of Different Stakeholders across the Three-Pronged Approach*

The engagement of various stakeholders, in a categorised (activity-based) manner is explained in Table 2 along with their specific roles, the challenges that were faced and how these challenges were overcome.

Whilst the demand-side and supply-side interventions were implemented side-by-side, however there is a degree of complementarity due to which both these aspects proved to be supportive to the Karula River initiative. A step-by-step process chart illustrating the implementation of activities under both types of interventions is available as Appendix D.

**Table 2.** Summary of roles, challenges and approach for this study.

| i. Supply Side Interventions | | | | | |
|---|---|---|---|---|---|
| **Roles of various stakeholders** | | | | **Challenges** | **Approach adopted to resolve** |
| *WWF-India* | *UPI&WRD* | *Farmers* | *District Authorities* | | |
| *Canal works:*<br>a. Canal de-siltation<br>b. Canal gate repair to ensure off-take of Designed Discharge throughout the canal<br>c. Repair and Maintenance of Outlet heads<br>d. Setting up hydrological monitoring system at Khanpur Minor and Karula River<br><br>*Canal-end to river-bank passage work:*<br>a. Along with farmers and department, identify most preferred route from tail-end of canal to Karula riverbank<br>b. Build consensus on the route and type of passage<br>c. Construction of passage in accordance with consensus | a. Permissions to carry out proposed work<br>b. Technical guidance in carrying out canal works, i.e., repair & maintenance<br>c. Technical supervision & monitoring of physical works<br>d. Regular maintenance and repair post-intervention<br><br>a. Convincing the farmers about passage formation, its route selection and support consensus building<br>b. Technical supervision of passage construction | a. Agree to become Ramganga Mitra<br>b. Participate in field surveys on canal for identification of works<br>c. Supervision of physical works on canals<br>d. Report any issue to the authorities and team<br><br>a. Agree on passage route<br>b. Convince fellow farmers for the initiative<br>c. Support while formation of passage<br>d. Maintain passage (as WUA function), beyond project duration | a. Support to carry out the work and provide contextual guidance, as required<br>b. facilitate Institutional synergies, i.e., to facilitate support from other departments for the purpose of the work<br><br>a. Facilitate to identify passage route & its formation<br>b. Convince command farmers to support this initiative | Canal system was in a dilapidated state, the passing of designed discharges from the head of the canal was not possible, plus several obstructions in the canal and therefore the tail-end area of canal generally remained un-fed Preferred route towards river Karula has a lot of encroachments by tail end farmers (mainly extension of farm boundaries). Therefore, sparing the space for passage route was one of the most challenging and complex tasks. | Complete rehabilitation of canal system was done, including—head-works repair, canal desilting, fixing of outlet head-pipes, Gauges repair & establishing new Gauge, clearing of obstructions etc.<br>Passage falls under tail-end village of command. Series of deliberations held with farmers and they were exposed to (i) the benefits of adopting improved practices and (ii) how they can contribute to a healthy Karula. Farmers got convinced to provide passage, but requested that most of the passage route should be underground and part of it should be on the edges of the farms to avoid damage to crops |

**Table 2.** *Cont.*

| | | ii. **Demand Side Interventions** | | | | |
|---|---|---|---|---|---|---|

| **Roles of various stakeholders** | | | | | Challenges | Approach adopted to resolve |
|---|---|---|---|---|---|---|
| *WWF-India* | *UPI &WRD* | *Farmers* | *Extension Agencies (Agriculture Science Center)* | | | |
| a. Building the capacity of the farmers towards Better Management Practices (BMPs) in irrigation & agriculture for Sugarcane <br> b. Demonstration of BMPs (Better Management Practices) & PoPs (Package of Practices) with farmers | Cropping pattern vis-a-vis irrigation water delivery information, with respect to various reaches of the Khanpur Minor canal | a. Agree to this initiative <br> b. Participate in trainings and exposure <br> c. Willingness to demonstrate BMPs & Package of Practices (PoPs) on their farms <br> d. Implementation of BMPs & PoPs on their farms | a. Progressive farming techniques <br> b. Support in development of Package of Practices (PoPs) <br> c. Knowledge Exchange, including exposure visits and on-farm sessions | | Sugarcane crop & flood-based irrigation is predominant in the region. Equitable distribution of water was a challenge. The situation aggravated by dilapidated state of canal & excess water being used by head-reach farmers leaving little for tail-enders. <br> Surface water irrigation is 100% subsidized for farmer's welfare, so there was no economic incentive to use less water | Being a cash crop, the recommendation for switching from sugarcane to another crop was deliberately not attempted. Therefore, the focus remained on improving the irrigation practices. The trench irrigation practice was introduced. Trench technique has not only resulted in reduction of canal water use but also reduced groundwater withdrawal, which certainly reduced input cost. In parallel, the farmers were sensitized for their role in reviving river Karula. |

| | | iii. **Institutional strengthening (including constitution of Khanpur Minor Water Users Association)** | | | | |
|---|---|---|---|---|---|---|

| **Roles of various stakeholders** | | | | | Challenges | Approach adopted to resolve |
|---|---|---|---|---|---|---|
| *WWF-India* | *UPI & WRD* | *Farmers* | *District Authorities* | | | |
| a. Guide, support and facilitate the process for constitution of Water Users Association (WUA), including –election process, voter list preparation & voter's validation <br> b. Trainings, Knowledge Exchange and Exposure Visits of command farmer's to active WUAs in & outside the state | a. Lead and coordinate the process for constitution of Khanpur WUA with 'Government of Uttar Pradesh' <br> b. Conduction of elections <br> c. Notify results & WUA constituted | a. Khanpur Minor WUA constitution process <br> b. Participate in the capacity building initiatives, including—training, exposure etc. | Facilitate and support the WUA election process | | Although the State Government promulgated UP Participatory Irrigation Management Act' 2009; but the process (farmer's awareness, Voter-List preparation & its validation, election schedule etc.) for WUA formation was time-taking | Series of awareness and training programmes were conducted. National & state-level exposure visits to successful WUAs were organised. The Voter List preparation and validation was facilitated. Khanpur Minor WUA is at place now. |

*2.4. Farmer Surveys—Approach and Methodology*

With an objective to assess the impact of the Karula river initiative on the farmers with respect to (i) on-farm water management and water savings and (ii) agricultural productivity and economic value of produce per unit of area, a detailed questionnaire (Appendix E) was developed. Based on this questionnaire, twelve farmer surveys were conducted jointly by some of the authors between 2018–2019 (sugarcane cropping season). For these farmer surveys, the sample selection was done from all the three reaches of Khanpur Minor canal, i.e., 2 farmers from each of the canal reach, i.e., head, middle and tail. The identification of the head, middle and tail end of the canal is done by dividing the total length of the canal into three equal parts. Within the reach the selection of farmer for survey was random. The farms where intervention (having BMPs) was made were noted as 'Demonstration-farms' and the ones with usual agricultural practices (without BMPs) were named as 'Control-farms'. It was noted while selecting the control farm, that both the control and demonstration farms belonged to similar specifications, except for sowing methods (with trencher and without trencher). Along with the field visit to all farms, detailed interactions based on the agreed questionnaire were conducted. Among all the command area farms, six sample demo plots (two each from head, middle and tail reaches of the canal) and correspondingly six control plots were selected to assess the impact and benefits of these interventions. Both the demonstration farms and control farms were geo-tagged, and their locations can be seen on the canal command area in Appendix F.

The farmer surveys were conducted through a combined approach, i.e., field-level measurements and 'farmer recall' method, this echo similar approach noted by Barton and Taron 2010 [16], while conducting representative farm surveys in the irrigation command areas in Tungabhadra River Basin, India.

There is a body of literature that talks about farmer-recall method as one of the means for conducting irrigation and agricultural surveys, especially in the absence of precise measuring and monitoring support. The analysis by Beegle et al., 2011 [17], as part of the work in three African countries, shows little evidence of recall bias impacting agriculture data quality at farm-level. They noted that the results of their work allay some concerns about the quality of some types of agricultural data collected through recall over lengthy periods. On the other hand, Wollburg et al., 2020 [18] find that, the recall length has a significant impact on reported outcomes in all areas of interest in agriculture surveys and analysis. They therefore suggested that, to reduce the risk of recall error and to improve the quality of key variables in agricultural surveys, shorter recall periods can be one of the solutions.

The authors, therefore, collected the information from the farmers during different stages of sugarcane crop, i.e., during land preparation, sowing, input applications and harvesting. Whilst multiple visits and interactions could be resource and time intensive; but, since the Karula river initiative has been a 4-year one and the team happened to visit field numerous times, which made it possible for the team to visit the farms and have discussions with the farmers during different phases of the crop cycle. This aspect is in alignment with the suggestions made in previous studies and research (Wollburg et al., 2021 [18], Beegle et al., 2011 [17], Barton and Taron 2010 [16]). Besides the farmer surveys, for the purpose of validation of information related to water application at sample farms, the team also measured the discharge and water levels in the field channels and farms.

The hydrological observations at Khanpur Minor were carried out through monitoring of gauge levels, active channel width and velocity to calculate discharges, which were used for water accounting. The observed discharge data is not available for the Karula river, as there is no monitoring station on this small river. It therefore becomes imperative to establish baselines which could later be utilised for comparison with the volume of saved water from irrigation discharged in the Karula river to improve its health.

The water used for sugarcane irrigation, both in demo and control fields, was compared with its ideal (theoretical) requirement. The actual discharge from tube wells with a 4-inch delivery pipe to the irrigation channel was measured at the site, using area velocity

method and volumetric measurement. On this basis, an average discharge of 0.014 cubic meter/second (40 litres per second) was adopted. A primary survey was conducted to gather information regarding the actual running time of tube-wells in demo and control plots for each irrigation/season. The volume of water applied in a field was calculated by multiplying the discharge with water application time. The irrigation water depth applied to a plot was calculated by dividing the total volume of water applied by the area of the plot. The ideal (theoretical) crop water requirement is as per FAO (Food and Agriculture Organization) norms using CROPWAT—a tool for calculating crop water requirements. The meteorological data of the nearest climatological station (Bareilly) was used. The rainfall data of the district Bijnor was taken, and the value of crop coefficient "Kc" was taken from guidelines issued by CWC (Central Water Commission, Government of India) in 1984 (Technical Series 2: A Guide for Estimating Irrigation Water Requirement, Ministry of Irrigation, Water Management Division, New Delhi, May 1984 [19]. The theoretical irrigation water depth for sugarcane crop computed using FAO's CROPWAT Program is calculated as 67.6 cm, including the 25% leaching requirement. Against this norm, the current irrigation water depth in control plots (without trench method) was calculated as 87.6 cm. The irrigation water depth in demo plots (with trench method) was calculated as 72.3 cm.

*2.5. Institutional Strengthening*

The state of Uttar Pradesh promulgated the Uttar Pradesh Participatory Irrigation Management Act in the year 2009 and since then the constitution of Water Users Associations (WUAs) at canal systems has been underway in a phased manner. So far, this work has been done in project areas of the Uttar Pradesh Water Sector Restructuring Project (funded by the World Bank). Hence, WUA formation in this area (Khanpur Minor, around the Karula river) had not begun. Under the Karula initiative, WUA was considered as an appropriate participatory institutional mechanism to sustain and take forward this initiative.

Work towards formation of WUA in the Khanpur Minor command area has been underway since 2018, with a series of awareness, sensitization and training programmes being conducted to build the capacity of farmers regarding WUA functioning, and its roles and responsibilities as per the Uttar Pradesh Participatory Irrigation Management Act, 2009 (UP PIM Act 2009). Exposure trips of farmers from Khanpur minor command area to successful WUAs in the state and outside the state have also been conducted. This way, a strong momentum was generated in favour of constituting the WUA and a critical mass of experts and vigilant farmers was readied to support the affairs of the WUA. Finally, in February 2021, the Khanpur Minor WUA was constituted with the unanimous election of its governing board members. The Khanpur Minor WUA was constituted following the provisions of the UP PIM Act 2009. The unanimous election results indicated the overall positivity amongst the command farmers towards the initiative as well as the institutional setup.

## 3. Results

This section discusses the findings of a sample survey of farms at all reaches of the Khanpur Minor canal, i.e., head, middle and tail reaches of the canal. Farmers from both typologies of farms, i.e., where interventions are being carried out, and where agriculture is still being practised in a traditional manner, were interviewed. The data from these interviews were analysed and the results are presented in this section.

This section essentially narrates the following:

1. Water savings at farm level
2. Flows restored in the Karula river
3. Change in sugarcane productivity
4. Economic implications for the farmers and crop-water productivity

### 3.1. Water Savings at Farm Level

The sugarcane crop raised using traditional practices (primarily, flood irrigation) consumed more water, whereas the crop raised using Better Management Practices (BMPs), including trench-based technique, consumed less water. The analysis of data shows average water savings to the tune of 17.4% using the trench method of sowing, (with the range between 40% and 10%) as shown in Figure 4. The saving of water can be attributed to the larger spacing among cane rows in the trench method.

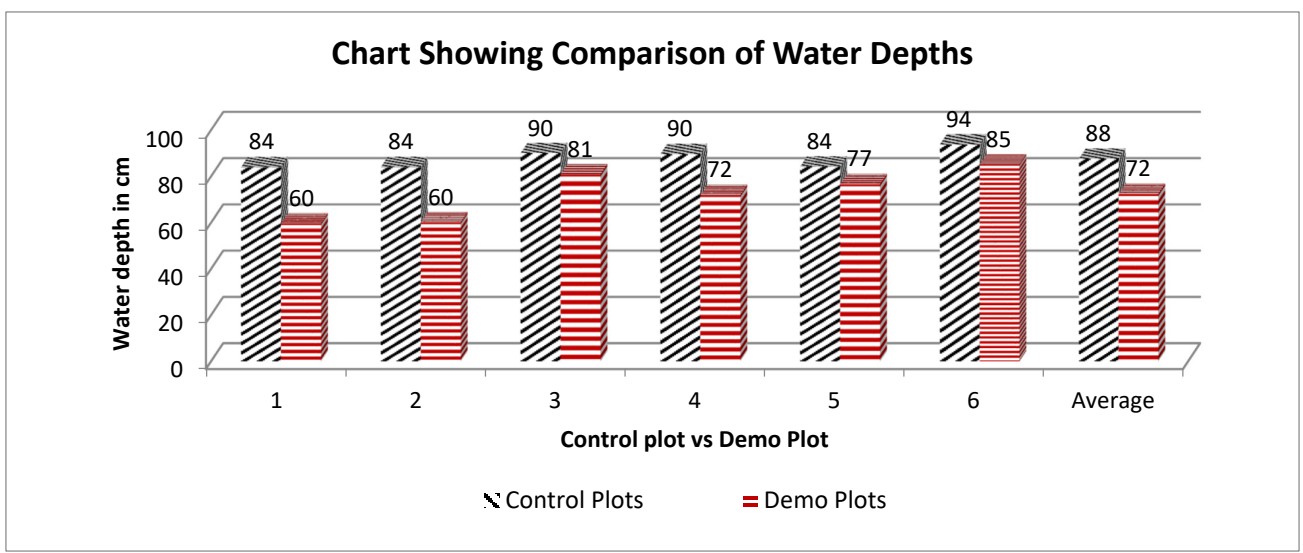

**Figure 4.** A comparison of irrigation water depths applied in control (where traditional way of irrigation is practiced) and demo (where trench-based irrigation is practiced) plots.

### 3.2. Flows Restored into the Karula River

Now, after the rehabilitation, the irrigation system is fully functional whenever the Khanpur Minor Canal gets water as per the roster (which is the mechanism of irrigation scheduling, that defines the date and time of water distribution for various canals within a system, in a turn-by-turn fashion) issued by the UPI & WRD. The canal system is run as per the roster. The saved water from the canal is now released into the Karula river through the passage. The Khanpur Minor Canal generally runs for 6–8 months in a year (depending upon water availability in the reservoir and irrigation water demand by the command farmers). From May 2019 until June 2021, the Khanpur Minor canal, through the passage, discharged a total of 62.55 million litres of water saved from irrigation to the Karula river across 67 days from May 2019 to June 2021. The discharge from the tail end of the Khanpur canal into the Karula river within this period ranged from 0.0033–0.022 cumec (3.4–22.6 litres per second), with an average flow rate of 0.011 cumec (11.9 litres per second), which is 11% of the 'designed discharge' of Khanpur Minor canal. Figure 5 shows the temporal variation in saved water discharged into the Karula river since May 2019.

Figure 6 gives an idea of minimum flows in Karula River and average discharge in Khanpur Minor vis-à-vis number of days in respective months, the saved water flown into the river.

From Figure 1, which shows the flow duration curve, it can be inferred that, at 90% dependability (leanest flows), about 0.085 cubic meter/second (85 L/s) water is available, whereas minimum average flows of 0.10 cumec (102 L/s) are observed in the month of June in the Karula, near the tail end of the Khanpur canal. It is also evident here that the saved water from irrigation discharged into the river Karula accounts for 7% of minimum lean season flows. It can be seen that except during the monsoon months (June to October) saved water from irrigation is discharged into river Karula during all the lean season months. With further adoption of Better Management Practices in the remaining sugarcane area in

command and the scaling up of trench-based interventions, it is expected that more water will be contributed by the Khanpur command to the Karula River.

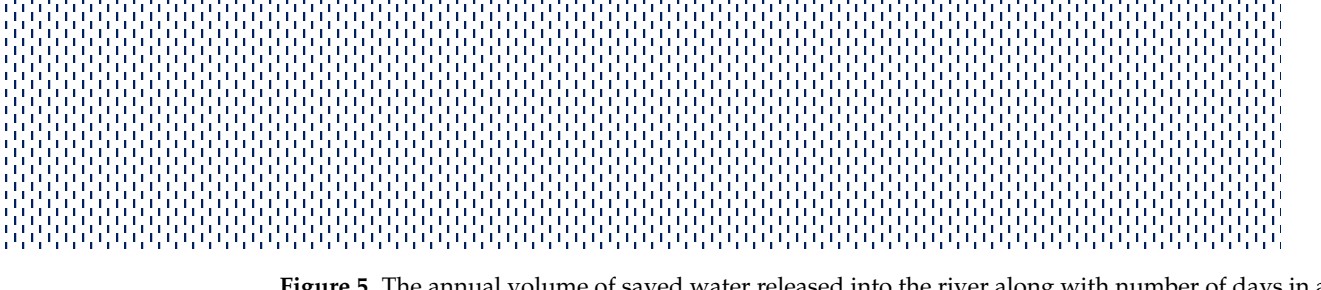

**Figure 5.** The annual volume of saved water released into the river along with number of days in a year when releases are made into the river.

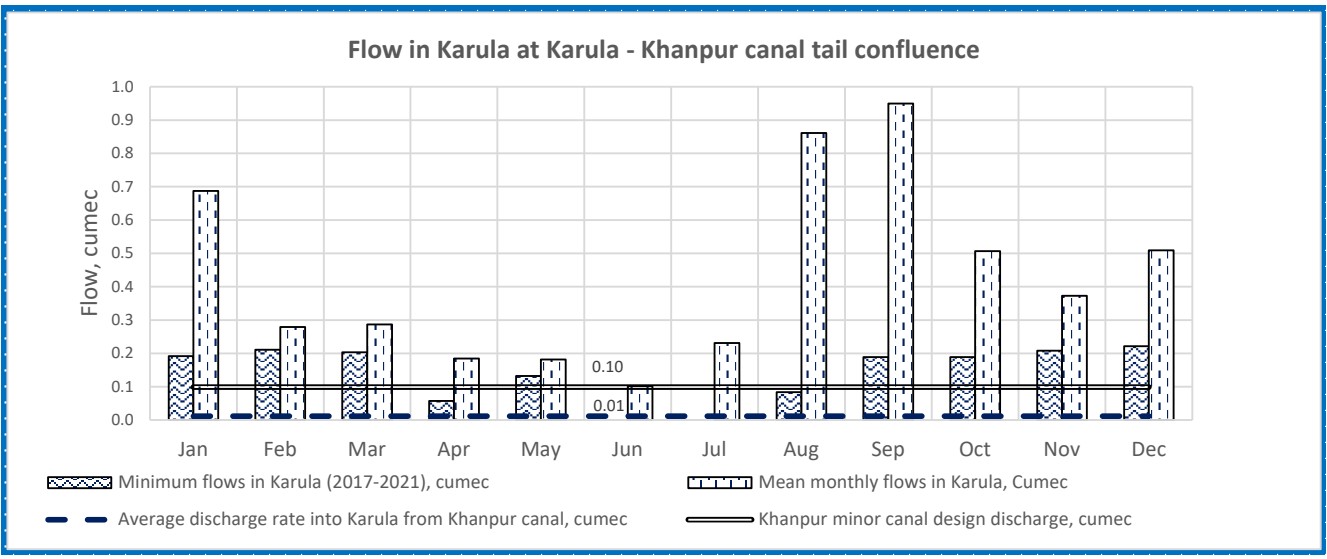

**Figure 6.** A snapshot of flow regime in Karula river and saved water from Khanpur Minor canal being discharged into Karula river.

*3.3. Changes in Sugarcane Productivity*

The data around sugarcane yield per unit area was discussed with the farmers. The figures around changes in yield (reported by the farmers) vary, depending upon the level/degree of adoption/adherence to Better Management Practices suggested, in addition to the adoption of the trench-based practice by individual farmers. Therefore, there may be some variations in the outcome or productivity levels.

In this case, of the six farms sampled, the general average trend of agricultural productivity enhancement is about 23.8%, with the range between 34% and 19%; Figure 7 exhibits the degree of change in sugarcane productivity.

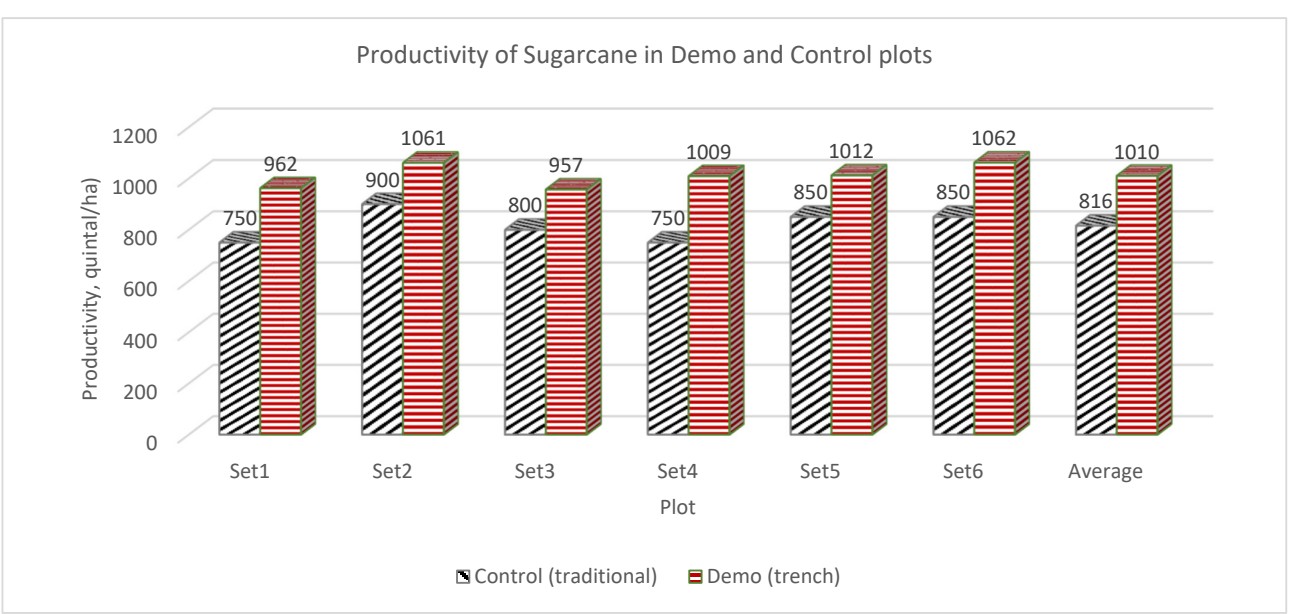

**Figure 7.** A comparison of productivity of sugarcane (Better Management Practices including trench vs traditional methods).

*3.4. Economic Implications for Farmers and Crop-Water Productivity*

Farmers have benefited in terms of earnings as well. The average income per unit hactare area, in comparison with control plot, is to the tune of Rs. 117,000/ha, whereas the range is Rs. 162,039/ha to Rs. 91,884/ha (Figure 8).

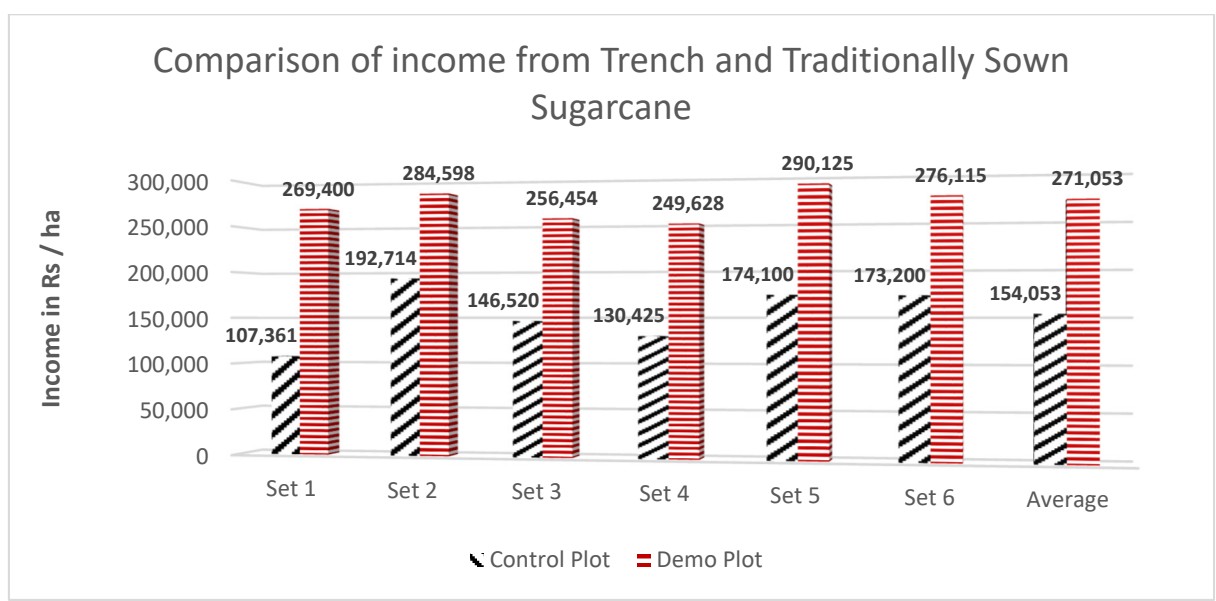

**Figure 8.** A comparison of income per hectare (Better Management Practices including trench vs traditional method).

Income per unit of water consumed (irrigation applied) was enhanced by 117 % (on average) in farms using BMPs than a traditionally sown farm (Figure 9). A farmer, on average, gets an additional income of Rs. 20.60 on every cubic metre of irrigation water used in the trench method. This is mainly due to a reduction in input costs (less fertiliser/pesticide, fuel etc.) and an increase in yield and higher returns.

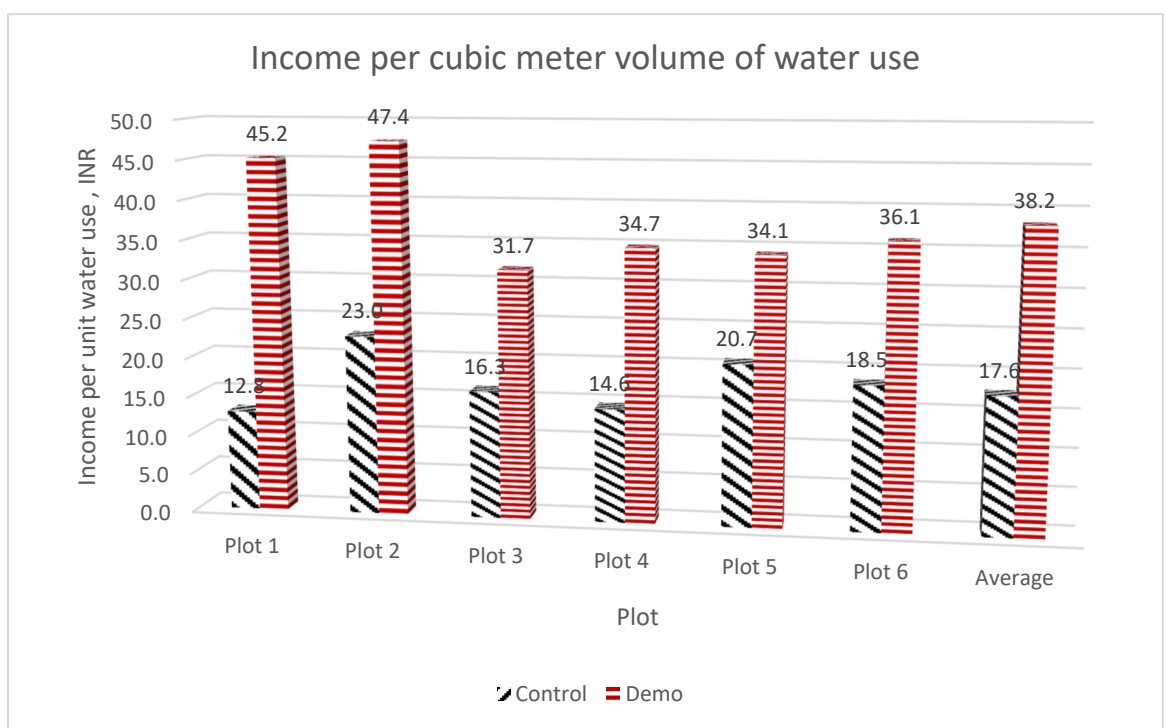

**Figure 9.** A comparison of income per cubic metre of water use (Better Management Practices including trench vs traditional methods).

The productivity per unit area may be attributed to the spacing between rows, which allows better aeration and provides space to grow freely, which results in cane plants of larger circumference and height, and weighing more, with greater sugar content. The per unit less water consumption may also be attributed to the heavier cane, providing greater yield of more value—with less water used.

Besides changes in sugarcane productivity and saving in irrigation water, the trench method offers opportunities to the farmers to grow a second crop in the sugarcane fields, simultaneously, between the ridges. Most of the farmers grow mustard or black-gram (*urad*) as an additional crop. These crops are not provided additional irrigation as their less water requirement is easily met with the soil moisture regime of the sugarcane crop. Farmers can use the additional crop for their consumption as well as to gain extra income from it. It has been calculated from demo farm data that the average income of multi-cropped sugarcane fields is around 20% higher (with the range between 15% and about 26%) than the single sugarcane crop sown with Better Management Practices, including the trench method, as shown in Figure 10.

The secondary crop, on average, contributes to around 17% (with the range between 13% and about 20%) of the total income of trench method sugarcane cultivation with multi-cropping. (Figure 11).

If the secondary crops had been sown alone, it would have consumed 15 cm irrigation water depth per hectare (assuming 50% area covered in sugarcane field is by the secondary crops, which consumes 30 cm water for maturity). This is totally saved by the irrigation water provided to the sugarcane. The total water requirement of both crops, if each crop is sown alone, comes to 87 cm (72cm + 15cm). Thus, the saving of 15 cm water of 87 cm is 17.2%. The sugarcane crop raised using trencher tool already has a saving of 17.4% over traditional sowing, hence the multi-cropping scenario offers total water saving of 34.6% over the traditional raising of sugarcane crop.

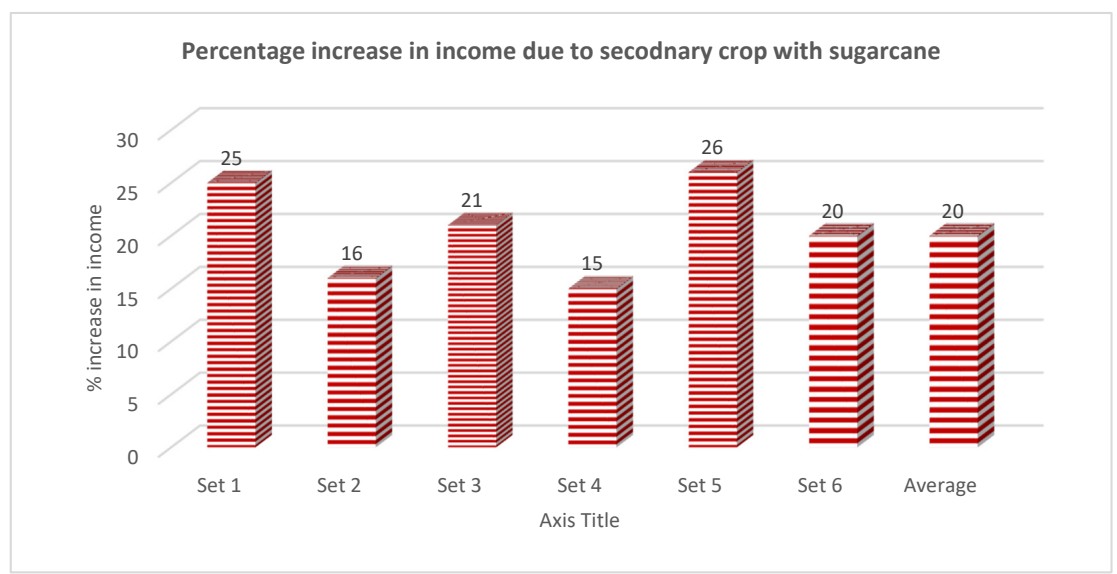

**Figure 10.** The percentage increase in income due to secondary crop with sugarcane.

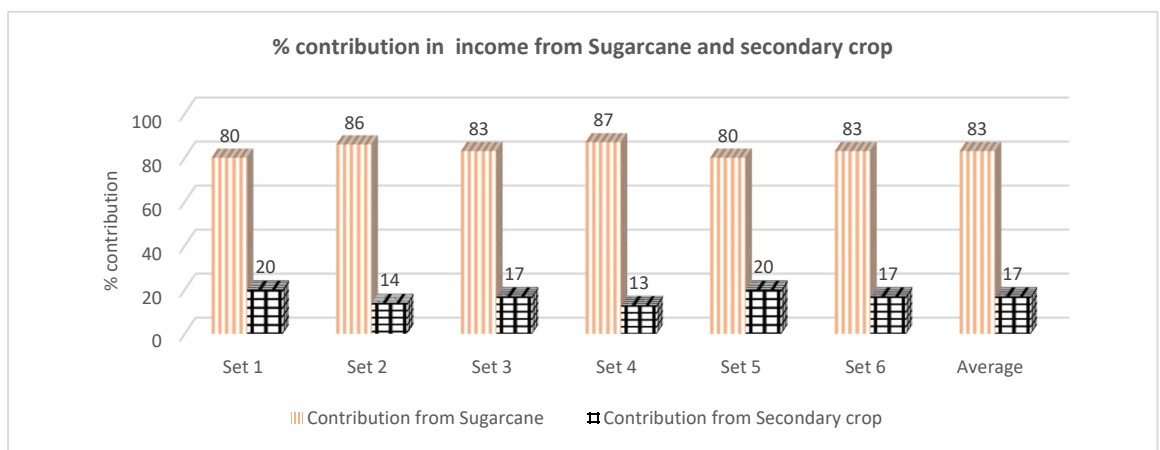

**Figure 11.** The percentage contribution of sugarcane and secondary crops in total income.

From the river conservation and water management perspective, the major outcome and impact of this initiative is the water savings from irrigation and release of that water into the river Karula through the passage. There are two sets of calculations—total water savings at farm level in view of using Better Management Practices (BMPs), including trench-based sugarcane farming, and actual water discharge data (from the gauge near the riverbank on the passage). These calculations are shown in Table 3.

**Table 3.** The current gains due to pilot project interventions.

| Water Savings from farm—Unit Area (in $m^3$/hectare) | Potential Water Saving If Trench-Based Sugarcane Adopted in All Farms in Khanpur Command ($m^3$) | Water Released into Karula River from Passage (in $m^3$) [Observed Data] |
|---|---|---|
| 1570 | 246,490 (from about 157 ha) | 62,550 |

Water saved to the tune of 62,550 cubic metres (25% of potential water savings) has found its way into the Karula river, thereby enhancing its flows. There are substantial conveyance (seepage) losses and unaccounted withdrawals, which has significantly reduced the overall volume of actual water released into Karula river. However, this means that there is an opportunity to bridge this inefficiency gap, so that the net gains can be enhanced.

## 4. Discussion

Globally, there are several initiatives, through which environmental gains are being tried and tested while implementing irrigation water use efficiency. Various researchers, over the period of time, have reviewed these initiatives.

Qureshi et al., 2010 [20] reviewed two key incentive policies to acquire water for e-flows for a part of the Murray-Darling Basin (MDB) in Australia. One policy consists of paying irrigators and water delivery firms to make capital and management investments that improve on farm irrigation and water-conveyance; the other policy consists of having the government buy water from irrigators on the active MDB water market. The result from their study shows that, the first option leads to relatively larger return flows reduction—which is not a welcome outcome. On the other hand, the second option tends to induce significant irrigated land retirement with relatively large reductions in consumptive use and small reductions in return flow. Thus, making the second option more viable than the first one.

Koech R. and Langat P. 2018 [21] reviewed the advancements made towards improving irrigation water use efficiency (WUE), with a focus on irrigation in Australia but with some examples from other countries. This body of work has demonstrated that the adoption of water-efficient technologies has delivered water savings at the field scale, with some of the savings being released as e-flows. However, the net water saving at the basin scale is not always achievable. In fact, some studies have demonstrated that a net increase in water consumption, largely due to the reuse of the saved water leads to expansion of land under irrigation.

On the other hand, a study by Ward and Velazquez 2008 [22] from Rio Grande basin, found out that water conservation subsidies are unlikely to reduce water depletions by agriculture. This study suggested that accurate accounting and measurement of water use can help identify opportunities for water savings, increase water productivity, and improve the rationale for water allocation among uses. Other measures include reducing or converting nonbeneficial evaporation from soil to beneficial crop ET, restricting acreage or water use expansion in cropped areas, switching to lower water-consuming crops, or irrigating current crops at a deficit.

Koech R. and Langat P. 2018 [21] further concluded that an overall reduction of water consumption at the basin scale is likely to be achieved when water-efficient technologies are used in combination with other measures, such as provision of incentives for water conservation and regulations to limit water allocation.

Unlike some of the global contexts, the water markets and subsidies for 'high-end' canal modernization, which comprise of pipelines, precision irrigation, linings etc. are not in practice at the moment in the catchment of Karula river; therefore, the basic improvisations in irrigation and agricultural practices holds the key to secure water for environmental requirements. This has been the thought process behind the propositions that were made to the farmers in the beginning of Karula pilot.

The Karula pilot was envisaged as a unique context-specific initiative, but under the backdrop of a well-debated idea—whether efficient irrigation water use can actually aid flows enhancement into the rivers and ultimately support the maintenance of e-flows in the rivers. On the other hand, there were externalities, which had the potential to disrupt the aspired outcomes of this initiative. However, a carefully developed stakeholder-led initiative has begun to deliver on the stated objectives, i.e., enhancing the flows in the river Karula. Herein, there have been favourable changes in terms of water use requirement from the demand-side and an efficient irrigation canal system, which ensured reliable water supply to the farmers. This has led to the achievement of saving water meant for irrigation and its release into the Karula river, besides benefiting the farmers economically. It is a combination of both these aspects along with institutional building, that powered the Karula pilot.

The specific values of water released into the river Karula would remain a dynamic figure, as there are several associated and external factors that would influence this. Some of these key factors could be as follows:

a.    The quantum of water flows in the Khanpur Minor canal, which may vary depending upon

  ✓    Availability of water in the main/parent canal
  ✓    Irrigation demand by farmers within the Khanpur command area
  ✓    Unauthorised withdrawals from the Khanpur Minor canal

b.    State of maintenance of Khanpur Minor
c.    Rainfall in the local catchment
d.    Maintenance of passage structure

To sustain such an effort beyond the project duration is indeed a challenging ask, as there would be an apprehension that the situation would be back to business-as-usual once the external support is withdrawn. To overcome this challenge, the formation of the Water Users Association (WUA) as per the provision of the Uttar Pradesh Participatory Irrigation Management Act 2009 was facilitated. In February 2021, the Khanpur Minor WUA was constituted and the elections for Executive/Governing Body (comprising of President, Secretary, Treasurer and other office bearers) of the WUA were unanimous. This is indicative of positivity amongst command farmers about the institutional support for this initiative, besides bringing them permanent solutions to the operation and maintenance of the Khanpur Minor canal system.

Parallel to the efforts to form the WUA at the Khanpur Minor level, the capacity building of the farmers about roles, responsibilities and functions of WUA was done through training programmes, exposure visits to successful WUAs in the state and at the national level. This has helped in mobilising a "critical-mass", who is now ready to take up the affairs of the WUA. However, the WUA is only recently established and further support will be needed for it to become fully sustainable in financial and institutional terms.

The Karula initiative was planned in such a way that the process for enhancement of flows in the Karula river fits within the current mechanism of irrigation scheduling and allocations and does not overwhelmingly change existing farm practices. This would mean that the envisaged objective is likely to achieve partial success in terms of actually maintaining the e-flows for a river. Therefore, the initiative may not by itself achieve the full suite of e-flows requirements (locations, timing and quantity of flows) for the Karula river, but it certainly aids to enhance the flows in the river in times of need, like the lean season of November to June.

Some local factors that worked in favour of the Karula pilot were as follows:

a.    Farmers in this area largely grow sugarcane (a water intensive crop) and the produce is insured by the Central and State government through Fair and Remunerative Price (FRP) and State Advised Price (SAP). Additionally, according to Niti Aayog (serves as the apex public policy think tank of the Government of India), the sugar mills that buy sugarcane are mandated to purchase crops from farmers within a specified radius known as the Cane Reservation Area at the FRP, which serves as defined market linkage for this cash crop. The team was fully aware of this fact—due to the availability of water and assured purchase of produce by the government through sugar mills, farmers would not switch to another water-intensive crop, which is a general apprehension otherwise.

b.    There has been another concern that farmers may tend to increase areas under agriculture using water saved from the application of Better Management Practices (including trench use) in sugarcane farming. Nevertheless, the team still faced a situation where, since the Khanpur Minor canal did not feed all the farms in the middle to tail-end, saturation of the command area was bound to happen –once the demand-side and supply-side interventions were applied in the command area, the saved water in the head to middle reaches of the canal would be used by the

tail-enders. As this was well-understood since inception and there was no hurried and strict response from the team to ensure that the saved water fed immediately into the river, the team worked with the tail-end farmers and assured them that they could use the water from the canal as well as from the passage for irrigation (by adopting trench-based technique), while letting the remaining water discharge into the river. The tail-end farmers agreed, and this strategy worked well.

c.     The other consideration in the Karula pilot is the promotion of local and scalable ideas to manage the demand-side aspect and not really call for hi-tech, expensive means of pressure irrigation (drip and sprinkler), at least in the early phases of the project. The idea was not to introduce something totally new to the area, but to bring some of the improvisations that are rare but known amongst the progressive farmers in and around that district. However, at a later stage, a few farmers proposed the idea of demonstrating pressure irrigation techniques and the team agreed to facilitate these.

Various scientific studies have suggested that water from seepage through unlined canals recharges groundwater (Mirudhula K. 2014) [23] and helps build shallow aquifers that are generally used as a source for irrigation. Infiltration from the canals recharges the aquifer directly and partially compensates for water uptake from plants and evaporation (Arumi J.L. et al., 2009) [24]. The idea behind this project was to support conjunctive use and reduce overall water withdrawal (canal and groundwater for irrigation), combined with improved practices in irrigation and agriculture, which is likely to reduce the losses from evapotranspiration, a matter of further investigation.

The groundwater serves the function of discharging base-flows into the river, especially during lean season. It was observed that excess infiltration from the flood irrigation technique (earlier prevalent in the command area), though, may be recharging shallow aquifers to some extent, but would also be increasing the overall evapotranspiration (ET) losses. Post field interventions, the volume of canal water applied has reduced, which may affect infiltration, but will also reduce the overall groundwater abstractions, subsequently helping in stabilising groundwater levels in the long run and will continue to feed the river through base-flows. Following the interventions in the Khanpur Minor command area to reduce abstractions, increase efficiency, and connect the canal tail to the river, the water has a more direct route to the river which augments riverine flows in its leanest flows periods. However, there are larger river-groundwater interactions in play too, which impact the riverine baseflows. Precise and conclusive information in regard to the exact benefits to the river and to the catchment will need to be inferred through long-term hydrological and hydro-geological monitoring.

Initiatives like the Karula river pilot can influence larger irrigation systems, as in a general scenario, the tail-ends of irrigation canals (in gravity-based systems) are close to rivers and wetlands. The saved water from irrigation, if conveyed to these freshwater resources, is likely to aid improvement of flows in the rivers. Arriving at such a stage is a critical milestone for maintaining e-flows in a river, because the most important question for e-flows maintenance is where the water for e-flows will come from, especially in over-allocated river basins. The irrigation water use efficiency initiative, as that of Khanpur Minor, could theoretically be upscaled at the extent of the Karula basin—about 65% (625 sq. km.) of catchment area of the Karula river grows sugarcane (as depicted in Appendix B). The extrapolations show that there is a potential of saving about 68 million cubic metres of water from about 70% of sugarcane farms within the Karula catchment. Whilst all the sugarcane farms in the Karula catchment may not be supported by surface-irrigation facilities (that could have otherwise directly demonstrated enhancing flows in Karula); however, potentially lesser groundwater pumping in view of application of Better Management Practices would certainly benefit the aquifer and river from these savings. This is likely to contribute to river discharges through enhanced base-flows. Moreover, there are about 30 minor irrigation canals in the adjoining areas of Khanpur Minor and

these are all fed by the Ramganga Canal. If this initiative could be up-scaled in these irrigation sub-systems, then more water could be augmented into the Karula river.

Whilst the apprehension may be valid that even if the water from irrigation is saved, it may lead to "enhancing-area-under-irrigation' and/or push for 'adoption-of-more-water-intensive-crops", in certain circumstances, the Karula initiative has proved that a carefully designed participative programme can actually bear desired results in terms of enhanced flows. These are complex questions and the same are being debated by various researchers. Globally, many countries and regions are trying to address and overcome similar challenges. For instance, the European Union (EU) as part of its agri-environment measures (AEMs) 2017 [25] provide financial support for Member States to design and implement programmes and projects. AEMs are developed under the EU's Member State's Rural Development Programme. They are mandatory for national and regional administrations, but voluntary for farmers. Farmers who choose to go beyond the current basic requirements can claim payments for AEM. Each measure has to have a specific environmental objective, such as—the protection or enhancement of biodiversity, soil, water, landscape, or air quality, or climate change mitigation or adaptation. Approximately 25% of the EU's utilised agricultural area is under AES contracts with farmers, including organic farming.

With the passage of the Water Act 2007, the Murray–Darling Basin (MDB) in Australia is in the process of major policy reform. This reform process is multifaceted and is expected to be completed by 2024. Key aspects of the reforms include: (i) the setting of sustainable diversion limits (SDLs) that will determine the average annual levels of extractions of water from surface and groundwater at a basin and catchment scale; (ii) the purchase, until 2015 of water rights in the form of water access entitlements for environmental purposes; and (iii) the on-going use of subsidies for water infrastructure to increase both on-farm and off-farm water-use efficiency. The progress of this ambitious and complex programme has been reviewed by many researchers (including—Grafton and Wheeler 2018 [26], Williams et al., 2019 [27]). While reviewing the possible effects of water recovery on river flows in MDB, Williams et al., 2019 [27] emphasized, the critical need to comprehensively measure the effects on recoverable return flows of increased irrigation efficiency, as a result of water infrastructure subsidies. It was further advocated by them that, good public policy requires a halt to any further water infrastructure subsidies in the Murray-Darling Basin to increase irrigation efficiency until it can be scientifically determined by how much, if at all, whether such infrastructure subsidies increase net stream and river flows, and at what cost. While commenting on similar aspects, Grafton and Wheeler 2018 [26] pointed out that, buy-back is a cost-effective measure in comparison to subsidies.

Whilst these large scale reforms are underway with varying complexities, smaller initiatives like this one on Karula River have the potential to provide some helpful pointers to advance or broadbase the outcomes of such reforms. On the other hand, the Karula initiative demonstrates an alternative to promoting radical changes (suggesting newer cropping patterns or promoting pressure-irrigation in the early stages) in a short time span, without much rapport building with the stakeholders. It would be much more prudent to look for local solutions (trench-based sugarcane farming, other package of practices including application of bio-pesticides and bio-fertilizers) and promote them in the project area. Once the benefits for the farmers are proven, they will come forward to support other forthcoming propositions as well.

*Practical Implications of Karula River Initiative*

The Karula River initiative provides some insights on how relatively straightforward and inexpensive interventions, co-designed with farmers and other stakeholders, can support achievement of environmental and socio-economic objectives in contexts similar to that found in the Karula River. Some of the key elements and takeaway points from the Karula River initiative are:

a.   The objective of irrigation water saving was to enhance the flows in the river, besides reducing the chemical inputs and increasing the agricultural productivity and therefore farm income.

b.   No high-end irrigation techniques (like drip, sprinkler) were considered across the project phase and mere improvisations in existing irrigation practices were promoted and implemented.

c.   No absolute and radical canal modernization (concrete/brick-lining of the canals, pipelines etc.) was done and mere basic canal rehabilitation was undertaken.

d.   Basic water management, water depth and water discharge monitoring mechanisms were considered (physical gauges with calibrated sections, flow-meters) instead of sophisticated tools and equipment.

e.   Promotion of unified approach as an institution, i.e., Water Users Association, rather than a fragmented one, i.e., at individual farmer-level.

These points can be some of the critical insights for similar initiatives in other river basins. It is proposed that a gradual, inclusive, multistakeholder-led process hold the key to the success of the work.

## 5. Conclusions

As lessons learned from the Karula initiative, the following takeaway points are therefore made, which may not be conclusive for further replication of similar ideas, but are certainly key pointers for future considerations:

a.   Integrated approach: rather than merely looking at a single aspect, a holistic and comprehensive view works better. For instance, instead of simply working on demand-side aspects, both supply-side aspects and institutional strengthening were also taken-up and this helped to achieve the objective. In addition, engagement with all key stakeholders, including the irrigation department, district authorities, local agriculture science centres and farmers, was critical for a transformational change

b.   Equity and Ownership: a saturation of canal commanded area, in terms of access to irrigation water across the various ends of the canal (head-middle-tail) is a necessary and critical step in such exercises and therefore this should be acknowledged to get wholehearted support from the farmers across all reaches within the canal system. Such considerations also allow better buy-in and sense of ownership amongst the farmers in the entire canal command area

c.   Monitoring: the monitoring of the transformation is a critical aspect and if this is done in a joint fashion, it adds value not only for the initiative, but also better informs the stakeholders about the change that is in the offing

d.   Scalability: considering a unit for proof-of-concept that is scalable, is critical, as the demonstration at an optimum unit has far better potential of upscaling, and therefore mainstreaming

Going forward, the team is now aspiring to upscale this initiative to about 16,000 hectare of Culturable Command Area (CCA) in the state of Uttar Pradesh, where the Ganga water resources feed the irrigation canals. This three-year programme will explore new leads, ideas, challenges, and opportunities, which would be worth narrating to a wider audience for their information, understanding and uptake.

It is fully recognized that the rejuvenation of some of the world's most populated and contested river systems continues to remain a challenging task, if the tributaries, rivulets, and wetlands in such river basins are not considered. It is in this context that the Karula pilot initiative is a pointer for policymakers and water-managers for the future. It is hoped that initiatives of this sort will help in curbing water-scarcity and will ensure wiser use of this precious resource. Moreover, the overall local ecology is set to benefit in this process as well.

**Author Contributions:** N.K., S.B., A.M., R.B., P.K.S. and R.K.A. were core members of the implementation team for Karula River initiative; N.K. conceived the idea of the paper and prepared first

draft of the paper, plus subsequent revisions to the paper.; S.B. provided valuable guidance and inputs throughout the paper drafting; A.M. and R.B. provided required inputs and played critical role in shaping the paper, while providing field level information and graphical representation of the findings; P.K.S. and R.K.A. led the farmer surveys in the field and guided the project implementation team across the initiative, the key findings from the farmer survey have emerged out of their surveys; D.T. and C.L. provided valuable insights and overall guidance from the global context and helped in shaping the paper across review process. All authors have read and agreed to the published version of the manuscript.

**Funding:** This initiative was supported by HSBC Water Programme (2017–2022).

**Data Availability Statement:** Not applicable.

**Acknowledgments:** The authors are grateful to Ravi Singh, Secretary General & Chief Executive Officer of WWF India for his constant motivation and support. The authors are also thankful to Sejal Worah, Programme Director of WWF India for her continuous encouragement and for being the source of inspiration to take up unique and new initiatives, like the one for the Karula river. WWF India's work on e-flows has been possible because of active support and valuable contributions from various partners, who have been part of this journey. The partners include several technical, scientific and academic institutions, civil society organisations and individual experts from the country. These entities include the Indian Institute of Technology, Kanpur, Integrated Natural Resources Management Consultants, Indian Institute of Technology, Banaras Hindu University, Varanasi, H.N.B. Garhwal University, Srinagar, Central Inland Fisheries Research Institute, Allahabad and People's Science Institute, Dehradun. Besides these, key government institutions like the National Mission for Clean Ganga and Central Water Commission, Government of India, the Irrigation and Water Resources Department of state of Uttar Pradesh and concerned district administrations (Bijnor and Moradabad) also contributed to the work. The authors and the WWF India team are indebted to senior officials of the Uttar Pradesh Irrigation & Water Resources Department, including VK Rathi, AK Singh and RP Singh for being supportive of the idea of the Karula initiative and also for providing their valuable guidance and contextual knowledge to carry out the work. The authors also thank DP Singh and Naresh Kumar, the senior field officials of the department for extending all field-based support to the initiative. The local field team of WWF India, including Anar Singh Yadav and Deepak Kumar contributed to the Karula initiative in a big way and the authors thank them for their contributions. The authors also thank WWF India partners, Ravindra Kumar and DK Dudeja for their continuous guidance and support in carrying out the e-flows work. The authors thank scores of farmers from the command areas of Khanpur Minor and adjoining canals for supporting, adopting and demonstrating the irrigation water management approaches proposed by the team.

**Conflicts of Interest:** The authors declare no conflict of interest. The funders had no role in the design of the study; in the collection, analyses, or interpretation of data; in the writing of the manuscript; or in the decision to publish the results.

## Appendix A. Brief History of Environmental Flows Assessment and Implementation in India

<div style="border:1px solid #000; padding:1em;">

### Recent History of E-Flows Assessment and Implementation in India

In the Indian context, a consortium of Indian Institutes of Technology, along with other partners developed Ganga River Basin Management Plan (GRBMP). This group defined E-Flows as, 'a regime of flow in a river or stream that describes the temporal and spatial variation in quantity and quality of water required for freshwater as well as estuarine systems to perform their natural ecological functions (including sediment transport) and support the spiritual, cultural and livelihood activities that depend on these ecosystems'.

WWF-India (World Wide Fund for Nature—India) has also been working towards E-Flows assessment and implementation, testing an assessment methodology with a multidisciplinary team of experts from other institutions and demonstrating field level interventions with local stakeholders. There are several initiatives from the government, civil society and academia who are working towards securing Environmental Flows in the river systems in India (updated from Gopal 2013) [9]:

- Minimum Flows—stipulations by Central Water Commission, Govt. of India 1992
- Deliberations and recommendations around E-Flows in Indian rivers by National Institute of Technology and International Water Management institute in 2001
- E-Flows assessment by Water Quality Assessment Authority—Govt. of India 2003-07
- Macro-level broad E-Flows assessment for Indian rivers by International Water Management Institute 2006 [28]
- Upper Ganga E-Flows assessment by a multidisciplinary team & WWF-India 2008–2010
- Aquatic species-centric E-Flows assessment for Upper Ganga by Wildlife Institute of India 2010–11
- Hydrology-based E-Flows assessment for Upper Ganga by Alternate Hydro Energy Centre 2010–11
- E-Flows assessment by consortia of IITs for Himalayan stretch of river Ganga 2011 [29]. The initiative was part of development of GRBMP
- National Water Policy 2012 [30], which called for maintaining E-Flows in river systems
- E-Flows for river Ganga by a multidisciplinary team, led by WWF-India for Triveni Sangam, Prayagraj location, Kumbh 2013 (Tare Vinod et al. 2013) [31]
- E-Flows initiative for Ramganga 2013 (Kaushal Nitin, Babu Suresh, Mishra Arjit, O'Keeffe Jay 2018) [32] and continuing, led by WWF-India
- The Ministry of Environment, Forests & Climate Change (Government of India) in the standard Terms of Reference for conducting the Environmental Impact Assessment studies for proposed River Valley and Hydro Project stipulated seasonal percentage of E-Flows that are to be maintained
- National Mission for Clean Ganga Authority Notification 2016 [33] by Government of India to call for maintaining of E-Flows in Ganga (National Mission Clean Ganga Gazette Notification, Government of India)
- Ganga E-Flows Order 2018 [34] and Amendment 2019 [35] by Govt. of India, stipulating specific E-Flows values for Ganga river (E-Flows Gazette Order 2018 [34] & Amendment 2019 [35])
- A joint initiative to assess E-Flows in all major rivers of Uttar Pradesh is underway (2019–22) by Uttar Pradesh Water Management & Regulatory Commission, Uttar Pradesh State Water Resources Agency and World Wide Fund for Nature—India. Under this initiative, the E-Flows assessment is done for Sharda, Ghaghra (Saryu), Gomti, Rapti, Yamuna, Son, Gandak rivers and plus some additional sites on Ganga River (where E-Flows assessment was not done in the past). The purpose of this exercise is to inform the exercise on River Basin Management Plans for these respective rivers by the Govt. of Uttar Pradesh.

</div>

## Appendix B. Land-Use Map of Khanpur Canal Command Area

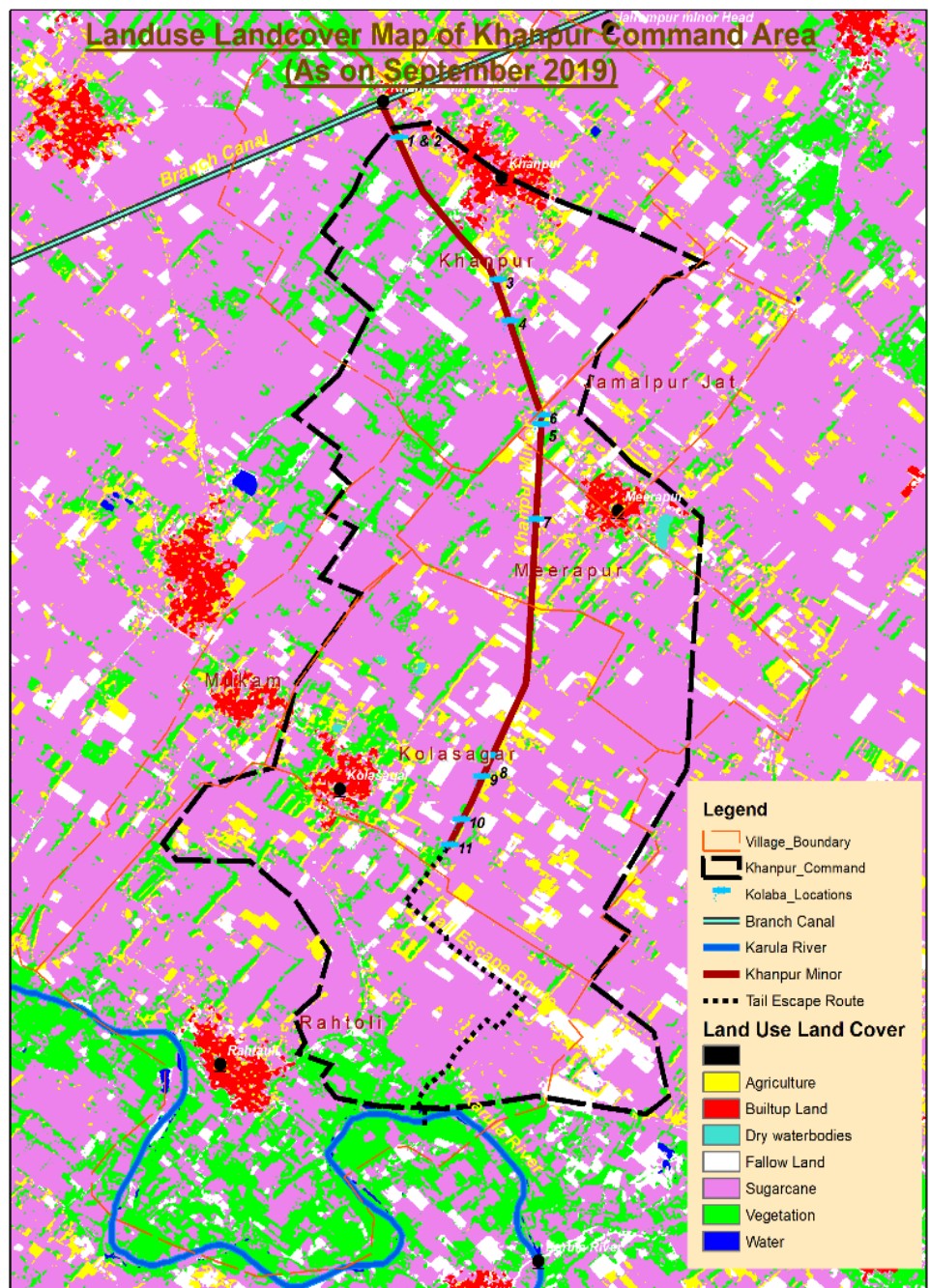

**Figure A1.** Land-use Map of Khanpur Minor canal command area.

## Appendix C. Land-Use & Land-Cover Map of Karula River Basin

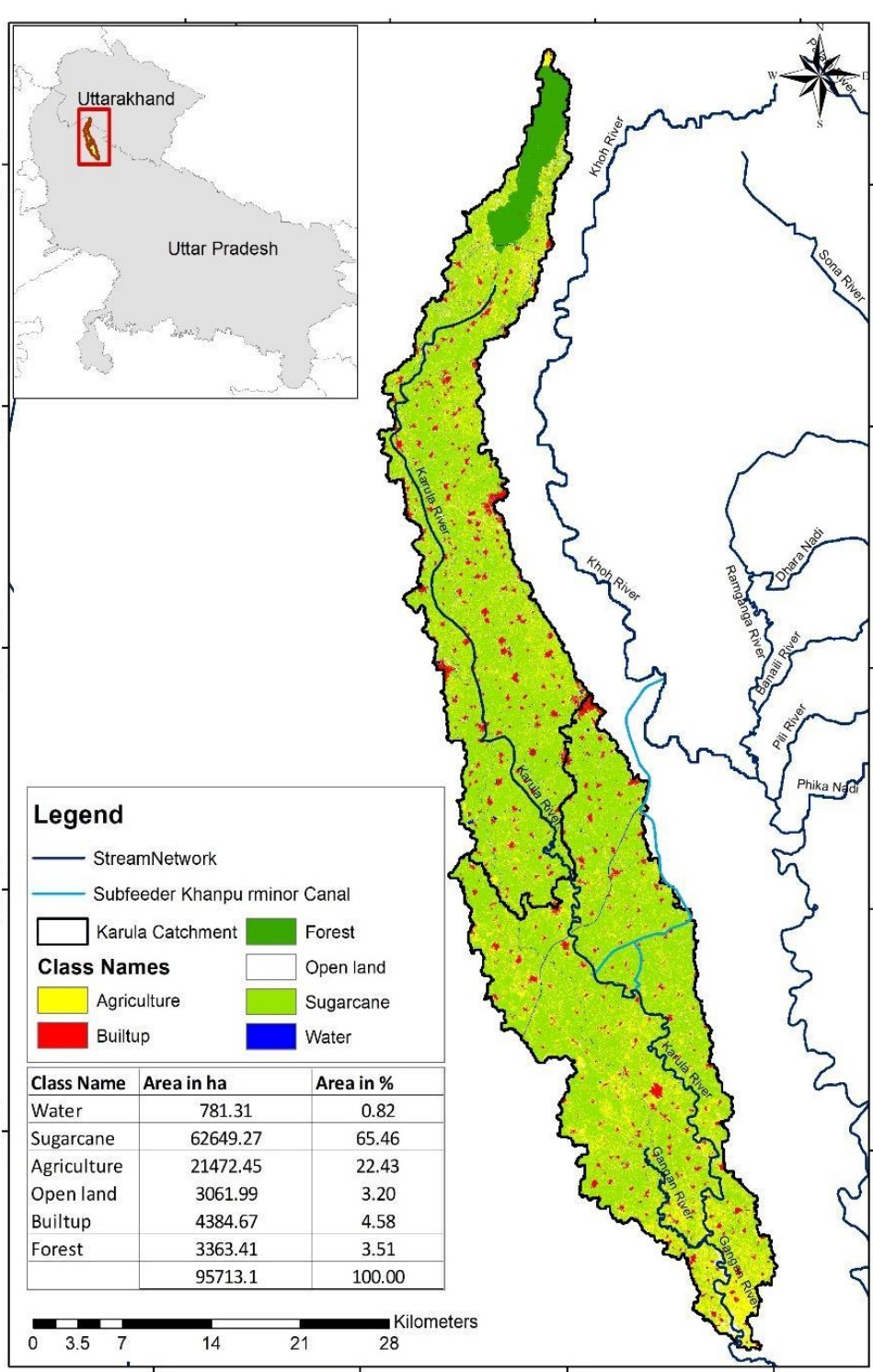

**Figure A2.** Land-use & Land-cover Map of Karula River Basin.

## Appendix D. Illustration of Combination of Supply-Side and Demand-Side Interventions Leading to Enhanced Flows in Karula

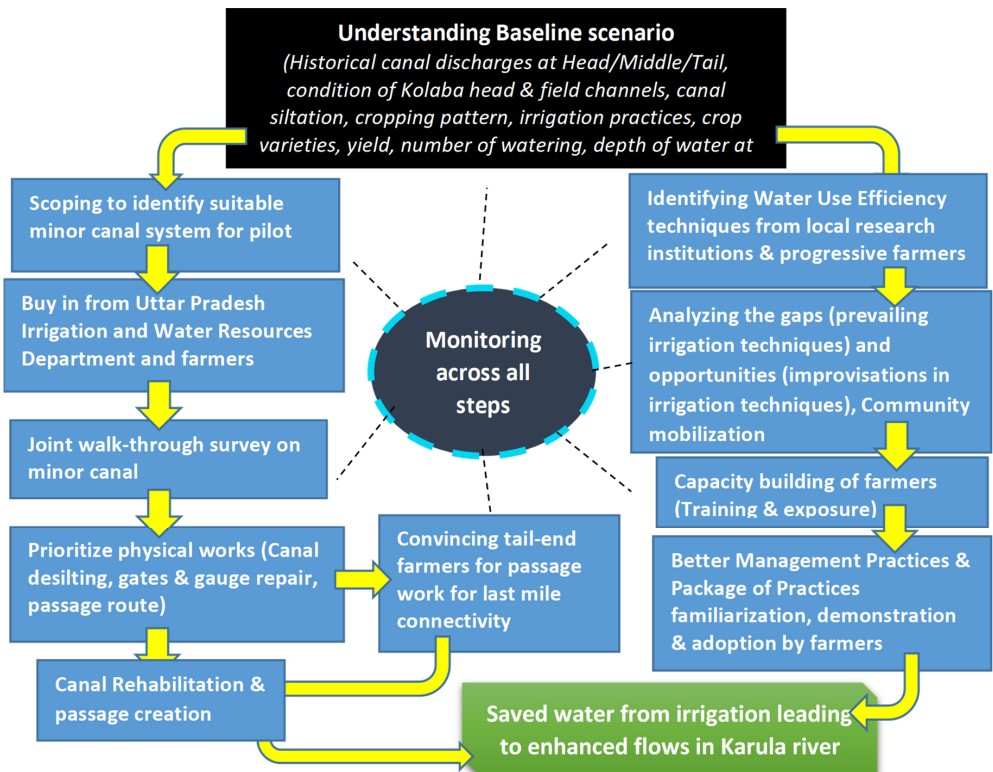

**Figure A3.** Illustration of Combination of Supply-Side and Demand-Side Interventions Leading to Enhanced Flows in Karula.

## Appendix E. Karula River Pilot—Farmer Surveys Questionnaire

Questionnaire for Joint Farmers Surveys Under Karula River Pilot
Objective

a.  to understand the agriculture and irrigation practices in both demonstration farms & control farms

b.  to ascertain the water-use at both categories of farms during watering and understand the variation in quantum of water that is used

c.  to understand the agricultural productivity and its economic value, while calculating the entire input costing; so that net economic gains can be assessed

1.  Basic details

    1.1  Date:
    1.2  Name of Farmer:
    1.3  Crop type:
    1.4  Farm size:
    1.5  Location on canal (H/M/T):
    1.6  Outlet Head Number

2.  Irrigation water application

    2.1  Name of crop:
    2.2  Method of Irrigation (flooding, basin, furrow etc.):
    2.3  Source of Irrigation (canal, tube well, well etc.):
    2.4  Total time of irrigation (calculated from irrigation time per watering and number of waterings per crop):
    2.5  Total water depth applied:

3.    Input details and costing

    3.1      Expense on seeds:

    3.2      Expense on labour (harrowing, ploughing, harvesting):

    3.3      Expense on compost:

    3.4      Expense on Fertilizers:

    3.5      Expense on Weedicides/pesticides:

4.    Productivity and economic value

    4.1      Sugarcane productivity per unit area:

    4.2      Other crop productivity per unit area:

    4.3      Market rate per quintal of sugarcane:

    4.4      Market rate per quintal of other crop:

**Appendix F. Khanpur Minor Command Area Map with Location of Control and Demo Farms**

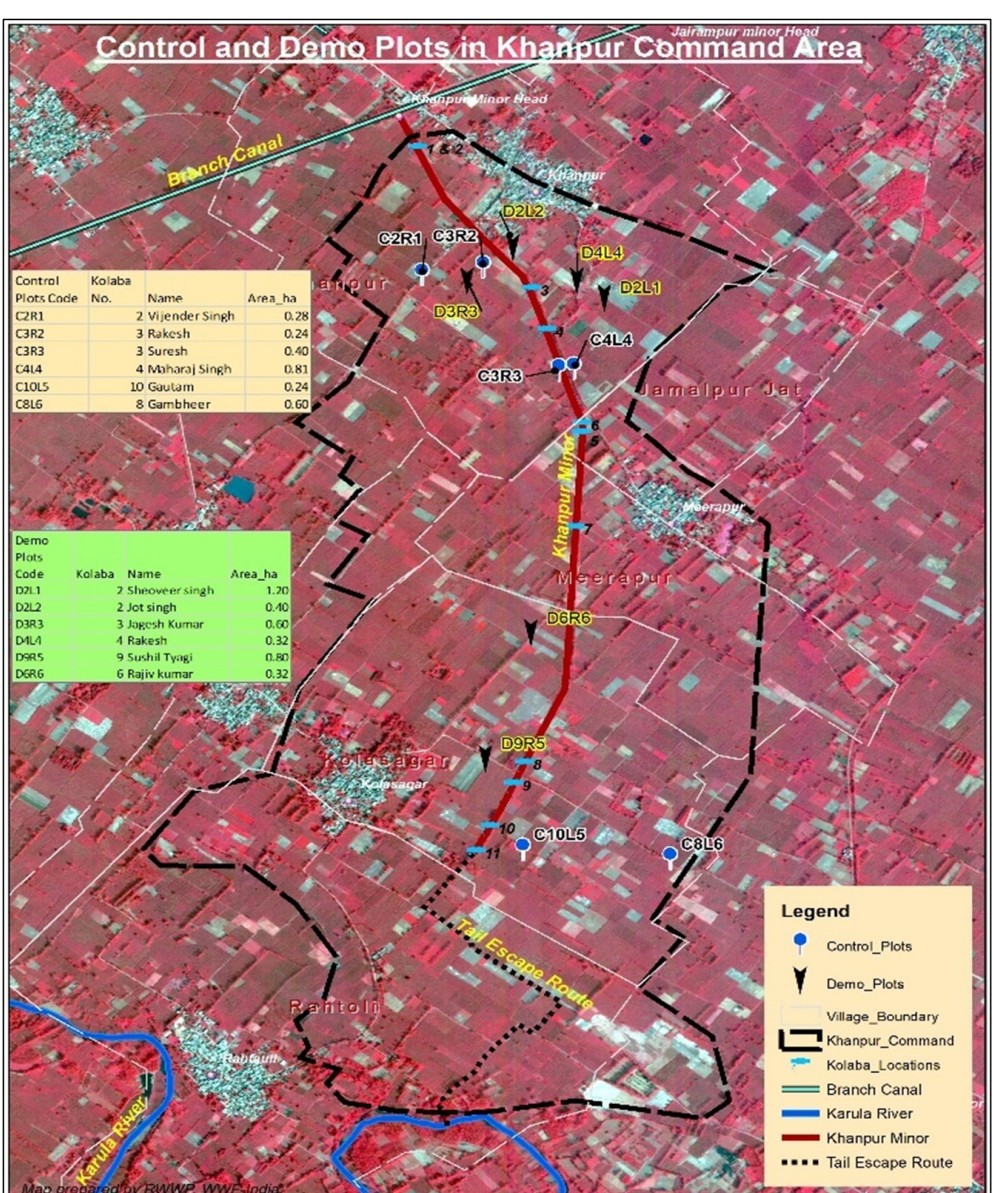

**Figure A4.** Khanpur Minor Command Area Map with Location of Control and Demonstration farms.

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
