# Peer review of "Securing Flows in the River Systems through Irrigation Water Use Efficiency—A Case Study from Karula River in the Ganga River System"

_water, doi:10.3390/w14182894_

Round 1

Reviewer 1 Report

The subject addressed in this manuscript is very relevant, considering that the problem of water scarcity and the waste resulting from agricultural activities is a worldwide problem.

However, the manuscript is poorly organized. The authors did it in a hurry, making "copy and paste" some report. The material has not been properly adapted to the structure of a journal article. Consequently, the manuscript is highly confusing for readers to read. On the other hand, the way the manuscript is written only has local/regional, not global, relevance. As the manuscript currently stands, it is in no condition to be published by "Water." However, due to the subject's relevance, I believe the manuscript should undergo major revision and be considered again for publication in "Water." Please, see the comments on the manuscript.

Author Response

Dear Reviewer,

We have carefully gone through all the comments and suggestions made by you, we thank you for reviewing the paper in detail. We find all the comments and suggestions well thought and we have addressed the same in the revised MS, being submitted now. As required the revised MS is submitted in track-change mode.
The key changes/modifications are as following:
1. All the referencing in the MS are done as 'End-notes'.- as suggested
2. The reference section, is now in the order of 'End-notes'- as suggested
3. Global experiences and context is detailed out in regard to similar initiatives across Europe and Australia- as suggested
4. Sub-sectioning is done for the required sections - as suggested
5. Key words ordering is revised - as suggested
6. Referencing of all Appendix
7. Moving the 'less-critical' text to Appendix
8. All graphs are replaced with the ones having better resolution
9. The metric unitary system is adopted
If needed a clean version can also be shared.

Reviewer 2 Report

This paper discusses the process, findings and lessons from a joint initiative involving farmers, the Uttar Pradesh state Irrigation and Water Resources Department, Bijnor District Administration and a conservation organization to enhance flows in a rivulet, called Karula River, which is part of the Ganga river system. In an attempt to describe the problem in detail, the authors have created a very comprehensive document that goes beyond the scope of standard contributions. From a formal point of view, the main aim is not clearly defined in the work. The manuscript contains numerous typos, stylistic issues, and some grammatical errors. The form of the listed graphs and images is not unified, the quality of which is very low. The authors provide a description of the graph, which is the same as the title, and the labels of the individual graph axes are not well processed. The graphs and legends therefore need improvement. The authors do not use the units of the SI system as well as the symbolism of their marking (e.g. cubic meters - m3). The discussion chapter, which lists a very limited number of cited sources with a similar theme, is very superficially handled. It is strongly recommended to add a subsection, 'practical implications of this study,' outlining the challenges in the current research, future work, and recommendations, before the conclusion. Conclusions should be more specific.

Author Response

  1. COMMENT No. 1: In an attempt to describe the problem in detail, the authors have created a very comprehensive document that goes beyond the scope of standard contributions. From a formal point of view, the main aim is not clearly defined in the work - some of the less critical content is moved to Appendix and required editing is done.  
  2. COMMENT No. 2: The manuscript contains numerous typos, stylistic issues, and some grammatical errors - every effort is made to rectify all such errors. 
  3. The form of the listed graphs and images is not unified, the quality of which is very low. The authors provide a description of the graph, which is the same as the title, and the labels of the individual graph axes are not well processed - All the graphs are reprocessed and reproduced. 
  4. The graphs and legends therefore need improvement. The authors do not use the units of the SI system as well as the symbolism of their marking (e.g. cubic meters - m3) - The SI system is adopted across the document. 
  5. . The discussion chapter, which lists a very limited number of cited sources with a similar theme, is very superficially handled. It is strongly recommended to add a subsection, 'practical implications of this study,' outlining the challenges in the current research, future work, and recommendations, before the conclusion. Conclusions should be more specific - a sub-section on 'practical implications of this study,' is inserted with key lessons.   

Round 2

Reviewer 1 Report

The manuscript has been significantly improved. Considering the relevance of the theme addressed, I am of the opinion that the manuscript has conditions to be accepted for publication after minor revision.

Suggestions for authors are listed in the manuscript. 

Author Response

Dear Reviewer,

Many thanks for 2nd round of the review. Baring couple of instances, all the suggestions and comments now stands addressed in the further revised MS, attached herewith. There is two comments for which we offer following explanation for clarity:

Line 82 (in PDF of commented MS): we are unable to find a much recent reference from same source.

Line 318 (in PDF of commented MS): Whilst, as per the UP PIM Act 2009, the constitution of WUAs is in process in many canal systems across Uttar Pradesh. But at the time of our project, this WUA constitution process did not start in Khanpur Minor, which was part of our work area, so the entire effort was started from the scratch. 
Hope this clarifies our position.  

We look forward to hear from you. 

Many thanks,

Nitin Kaushal 

Reviewer 2 Report

The revised version contains a number of modifications based on the opponent's comments and suggestions. I strongly draw attention to the necessity of revising the cited literature and unifying the methods of their citation according to the instructions for authors. After incorporating these changes, it is possible to accept a contribution for publication.

Author Response

Dear Reviewer,

Many thanks for all the comments and suggestions. 

We tried our best to address your comments and suggestions, mainly pertaining of proper referencing and citations. 

We are having a trouble in regard multiple referencing of one literature source; so while we hope that would be helped by the Journal (as we are unable to address it). There are 3 such references, so for the time being, we have manually given the same reference number as that of the original one. 

Hope this can be resolved at a later point in time.

Many thanks,

Nitin Kaushal 
